# Antibody–Drug Conjugate Made of Zoledronic Acid and the Anti-CD30 Brentuximab–Vedotin Exert Anti-Lymphoma and Immunostimulating Effects

**DOI:** 10.3390/cells13100862

**Published:** 2024-05-17

**Authors:** Feliciana Morelli, Serena Matis, Roberto Benelli, Laura Salvini, Maria Raffaella Zocchi, Alessandro Poggi

**Affiliations:** 1Molecular Oncology and Angiogenesis Unit, IRCCS Ospedale Policlinico San Martino, 16132 Genoa, Italy; feliciana.morelli23@gmail.com (F.M.); serena.matis@hsanmartino.it (S.M.); roberto.benelli@hsanmartino.it (R.B.); 2Fondazione Toscana Life Sciences, Technology Facilities and Mass Spectrometry Unit, 53100 Siena, Italy; l.salvini@toscanalifesciences.org; 3Division of Immunology, Transplants and Infectious Diseases, IRCCS San Raffaele Scientific Institute, 20132 Milan, Italy; marazocchi55@gmail.com

**Keywords:** Hodgkin lymphoma(s), Vδ2 T-cells, ADC, zoledronic acid, brentuximab, CD30

## Abstract

Relevant advances have been made in the management of relapsed/refractory (r/r) Hodgkin Lymphomas (HL) with the use of the anti-CD30 antibody–drug conjugate (ADC) brentuximab–vedotin (Bre–Ved). Unfortunately, most patients eventually progress despite the excellent response rates and tolerability. In this report, we describe an ADC composed of the aminobisphosphonate zoledronic acid (ZA) conjugated to Bre–Ved by binding the free amino groups of this antibody with the phosphoric group of ZA. Liquid chromatography–mass spectrometry, inductively coupled plasma–mass spectrometry, and matrix-assisted laser desorption ionization–mass spectrometry analyses confirmed the covalent linkage between the antibody and ZA. The novel ADC has been tested for its reactivity with the HL/CD30^+^ lymphoblastoid cell lines (KMH2, L428, L540, HS445, and RPMI6666), showing a better titration than native Bre–Ved. Once the HL-cells are entered, the ADC co-localizes with the lysosomal LAMP1 in the intracellular vesicles. Also, this ADC exerted a stronger anti-proliferative and pro-apoptotic (about one log fold) effect on HL-cell proliferation compared to the native antibody Bre–Ved. Eventually, Bre–Ved–ZA ADC, in contrast with the native antibody, can trigger the proliferation and activation of cytolytic activity of effector-memory Vδ2 T-lymphocytes against HL-cell lines. These findings may support the potential use of this ADC in the management of r/r HL.

## 1. Introduction

The involvement of γδT cells in the control of tumors, including leukemias and lymphomas, is confirmed [1,2,3]. In particular, γδT lymphocytes can recognize phosphoantigens (pAg) accumulated and presented by neoplastic cells [4,5,6,7]. Also, the antitumor activity of these lymphocytes, bearing at the cell surface the Fcγ receptor IIIa/CD16, can be triggered through the Fc portion of therapeutic antibody [7,8,9].

Different compounds, such as aminobisphosphonates (N-BPs), can stimulate the proliferation of the peripheral blood Vδ2 T-cell subset, increasing the number of circulating γδT-lymphocytes able to destroy tumor cells [5,6,7,8]. Among N-BPs, zoledronic acid (ZA) is a chemically stable analog of inorganic pyrophosphate able to interfere with the farnesyl pyrophosphate (FPP) and geranylgeranyl pyrophosphate (GGPP) synthase of the cholesterol synthesis pathway. This inhibition induces the intracellular accumulation of isopentenyl pyrophosphate (IPP); in turn, the IPP is presented by the cell-membrane butyrophilins such as BTN3A1 and BTN2A1 to the Vγ9Vδ2 T-cell receptor eliciting their preferential growth and activation [7,8,9,10]. The two mechanisms, immunostimulation and direct anticancer action, are the rationale for the use of different N-BPs in clinical trials [11,12,13].

We have reported that ZA can stimulate the expansion of Vδ2T-cells with anti-leukemic and lymphoma activity, possibly acting as a therapeutic tool to enhance conventional therapies [14,15]. In the specific case of Hodgkin lymphoma (HL), monoclonal antibodies (mAb) directed to the CD30 antigen expressed by tumor cells represent a key therapeutic tool, especially for treating relapsed/refractory (r/r) HL [16,17,18]. Indeed, the use of the anti-CD30 antibody brentuximab linked to the microtubule inhibitor vedotin leading to the antibody–drug conjugate (ADC) named brentuximab–vedotin (Bre–Ved) can evoke a strong anti-HL cytotoxic effect improving the chemotherapy [16,17,18]. Also, resistance is not rare, and in second-line treatments, survival advantage is often lower than expected and is reached only in combination with other antiblastic drugs [16,17,18]. The expression of CD30 on HL is finely regulated by the action of a disintegrin and metalloproteinase (ADAM) 10 activity, and Bre–Ved effects can be increased using specific ADAM10 inhibitors [15]. Thus, it is conceivable that the anti-CD30 ADC-mediated therapeutic effect could be enhanced by eliciting the generation of anti-lymphoma effector lymphocytes as well as using drug combinations [15]. It should be considered that although several ADCs have been synthesized for treating different types of cancers [19,20,21] exploiting the cytotoxic effect of a drug linked to an antibody, only a few are currently used due to incomplete or unsatisfactory clinical outcomes [19].

More recently, we have described a new type of ADC, based on the anti-EGFR therapeutic antibody Cetuximab, covalently linked to ZA exerting anticancer properties [22]. This ADC can trigger the expansion of Vδ2 T-cells in patients suffering from colorectal carcinoma that, in turn, can kill patient-derived organoids. This effect was mediated by both Vδ2 TCR triggering and antibody-dependent cellular cytotoxicity (ADCC) [22].

In this paper, we study the reactivity and functional properties of a new ADC, exploiting the immunostimulatory properties of ZA covalently linked to Bre–Ved. Bre–Ved targets CD30 and is used in the therapy of Hodgkin lymphoma (HL) due to the antiblastic effect of monomethyl-auristatine. In particular, we show that: 1. Bre–Ved–ZA ADC, covalently linking ZA to Bre–Ved, can be obtained through the generation of a phosporamidate. 2. Bre–Ved–ZA ADC maintains the same specificity but a stronger anti-lymphoma activity compared to the ZA-unconjugated Bre–Ved therapeutic antibody. 3. Bre–Ved–ZA ADC can trigger the expansion of Vδ2 T-cells with effector-memory phenotype and antitumor activity.

## 2. Materials and Methods

### 2.1. Synthesis and Chemical Characterization of Bre–Ved–ZA ADC

Bre–Ved (Adcetris^®^) was obtained as a leftover of the therapeutic preparation used for patients suffering from HL (Pharmacy Unit of IRCCS Ospedale Policlinico San Martino, Genoa, Italy) and zoledronic acid (ZA, MW 272.09) was purchased from Selleckchem (Houston, TX, USA). The anti-CD30 antibody Adcetris^®^ was dialyzed following the procedure described for linking the free phosphoric groups of DNA to peptides that have been coupled to ZA, as described in detail elsewhere [22,23,24,25]. Bre–Ved–ZA was then dialyzed with the Slide-A-Lyzer Cassette (Thermo Fisher, Waltham, MA, USA) at 10,000 MWCO for 24 h. The ADC was analyzed using matrix-assisted laser desorption/ionization–mass spectrometry (MALDI–MS, Cytiva Italy srl, Milan, Italy) to determine the covalent linkage between the antibody and ZA. The Bre–Ved and the corresponding Bre–Ved–ZA ADC were desalted by Spin Trap G25 (Cytiva Italy srl, Milan, Italy). A total of 2 μL of each compound was mixed with 2 μL of a saturated solution of s-DHB in 0.1% TFA in acetonitrile:water (50:50, *v*/*v*). The mixture was deposited on the MALDI plate and left to dry in the air. Successively, each spot was combined with 1 μL of the saturated matrix solution, allowing it to dry. A protein standard II calibration mixture (Bruker Daltonics, GmbH, Bremen, Germany) was used for calibration. UltrafleXtreme (Bruker Daltonics, GmbH) MALDI mass spectrometer was used to acquire the mass spectra over the mass range *m*/*z* 30–220 kDa, using linear mode (Figure 1).

Induction coupled plasma–MS (ICP–MS) was performed by Nanovex using the 8900 ICP-QQQ Analyzer (Agilent Technologies, Milan, Italy) to determine the amount of phosphorus (P) coupled to the antibody compared to the amount of sulfur (S) [22]. Also, the drug:antibody ratio of ZA and Ved to brentuximab was calculated. Briefly, the liquid chromatography–mass spectra (LiqC–MS) analysis of reduced Bre–Ved–ZA was performed using a high-resolution mass spectrometer Orbitrap Q-Exactive Plus (Thermo Fisher, Waltham, MA, USA) coupled with an Ultimate 3000 liquid chromatography system (Thermofisher Scientific). The chromatographic run was performed on a bioZen™ 3.6 µm Intact C4, 100× 2.1 mm column (Phenomenex srl, Bologna, Italy). The mass spectra were acquired in the 700–3000 *m/z* range in electrospray in positive-ion mode. The software BiopharmaFinder 2.1 was used for data elaboration. The antibody sample was prepared to estimate the average DAR of Bre–Ved–ZA, and the ADC was reduced with 100 mM DTT at 37 °C for 1 h. The resulting mixture composed of conjugated and native light chains and heavy chains was desalted using an Amicon-Ultra 0.5 mL, 10 MWCO (Merck Millipore, Darmstadt, Germany) and analyzed by LiqC–MS. No deglycosylation was performed before the analysis. The LiqC–MS analysis of the mixture resulting from the reduction of Bre–Ved–ZA resulted in the complex chromatogram reported in Figure 2.

To obtain information on all the molecular species formed from Bre–Ved–ZA, the raw data were compared with the amino acid sequence of brentuximab by Biopharma Finder 2.1 (Thermo Fisher). Vedotin and zoledronic acid were imposed as variable modifications. The results are summarized in Table 1. In the first column, the measured average mass (in daltons) for light chain (LC) or heavy chain (HC) species experimentally detected are shown. In the third and fourth columns, the sum of the intensity and the charge state distribution of the mass spectra are listed. The table is subdivided into two groups: the first regarding the LC and the second the HC. In the fourth column, the delta mass is listed. The last column reports the assignment for each molecular species.

### 2.2. Flow Cytometry and Confocal Microscopy

KMH2 or L428 (from pleural effusion) and L540 (from bone marrow) classical HL-cell lines (purchased and certified from DSMZ GmbH, Braunschweig, Germany), or the CD30^+^ lymphoblastoid RPMI6666 and HS445 cell lines (provided and certified by ATCC, Manassas, VA, USA) were incubated with serial dilutions (20–0.02 µg/mL/10^6^ cells) of Bre–Ved or Bre–Ved–ZA ADC, for 1 h at 4 °C, followed by the AlexaFluor647 antihuman Ig (AF647-α-hIg) labeling. Negative controls were stained by AF647-α-hIg only. Samples were analyzed by a Cytoflex S flow cytometer (Beckman–Coulter Inc., Brea, CA, USA), and results are expressed as log mean fluorescence intensity (MFI, arbitrary units, a.u.) or percentage of positive cells. For confocal microscopy, all HL-cell lines plated onto 96-well clear flat-bottomed black plates for imaging (Eppendorf, Hamburg, Germany) were incubated with 2 µg/mL Bre–Ved–ZA ADC for 1 h at RT, followed by AF647-α-hIg and SytoX Orange 200 nM (Thermo Fisher Scientific, Monza, Italy) staining. To detect Bre–Ved or Bre–Ved–ZA internalization, HL-cell lines were incubated at 37 °C for 24 h with the corresponding antibody (2.0 µg/mL), fixed with paraformaldehyde 1%, permeabilized with 1% Triton X-100, followed by AF647-α-hIg antiserum. Lysosomes were contemporarily identified by an anti-LAMP1 mAb (clone H4A3, Thermo Fisher Scientific) followed by isotype-specific AF488-α-mouse antiserum (Thermo Fisher). Cell samples were observed by confocal microscopy FV500 (Olympus Europe GMBH, Hamburg, Germany) and analyzed as described [14]. Results are shown in pseudocolor: blue (nuclei), red (anti-CD30 mAb), and green (LAMP1).

### 2.3. Evaluation of HL-Cell-Line Proliferation and Apoptosis

KMH2, L428, L540, HS455, or RPMI6666 (5 × 10^3^ cells/well) were tested with titrated amount of Bre–Ved or Bre–Ved–ZA (20, 2.0, 0.2 μg/mL) and cultured at 37 °C. The concentration of each antibody was determined with the QuantumMicroProtein kit from Euroclone (Milan, Italy) following the manufacturer’s instructions. On day 5, ATP content was determined using the CellTiter-Glo^®^ Luminescent Cell Viability Kit (Promega Italia Srl, Milan, Italy) following the manufacturer’s instructions. Luminescence was detected with the VICTORX5 multilabel plate reader (Perkin Elmer, Milan, Italy), expressed as luminescence arbitrary units (a.u) [14]. Images of proliferating HL-cells were taken at 24, 48, and 72 h with the CELLCYTE X^TM^ imaging recorder (Cytena, Breisgau, Germany); the cell area was calculated by image analysis using the CELLCYTE Studio software (https://www.cytena.com/new-cellcyte-x-software-january-2022/) and expressed as mm^2^ (a representative example for the software mask applied at 24 h is shown in Appendix A). The same samples were stained by the C.LIVE Tox green probe (20 nM, Cytena, Breisgau, Germany) to identify dead cells by the resulting fluorescent signal (expressed as arbitrary units: a.u./well), using the CELLCYTE Studio software.

Long-term proliferation was tested on the KMH2 cell line (5 × 10^4^ cells/well/mL medium, in 24-well plates, with or without Bre–Ved or Bre–Ved–ZA (20.0, 2.0, 0.2, 0.02, and 0.002 µg/mL serial dilutions)). Cultures were maintained in a 5% CO_2_ incubator at 37 °C, microscopically examined every day, and split when necessary. Cells were quantified in each culture condition on days 7, 14, and 21 by a LUNA-II^TM^ automated cell counter (Aligned Genetics, Inc., Logos Biosystems, Anyang-si Gyeonggi-do, Republic of Korea), harvesting 100 µL of culture medium from the cell suspension. Cell proliferation in each experimental condition was plotted as a fold increase of the initial amount at each time point, calculated by multiplying the number of cells counted for the number of 1-in-2 splits.

Apoptosis was evaluated at 72 h with the luciferase-based Caspase Glo-3/7 and Glo 9 3D Assays (Promega) following the manufacturers’ instructions. The output was read on a Luminometer (VictorX5, Perkin-Elmer Italia, SpA, Milan Italy), and results were expressed as luminescence units (RLU)/5 × 10^4^ cells. In other experiments, apoptotic cells were identified by annexin-V (Bender MedSystem GmBH, Vienna, Austria) labeling, as described [14]. Apoptotic or necrotic cells were differentiated by flow cytometry (Cytoflex S flow cytometer, Beckman–Coulter) after propidium iodide (PI) staining of non-permeabilized cells. At least 10^4^ cells per sample were analyzed, and apoptotic cells were identified as annexin-V^+^PI^+^ cells. Apoptotic cells were also analyzed by confocal microscopy in a bright field or after staining with Syto16 100 nM to evidence nuclei fragmentation [14].

### 2.4. Vδ2 T-Cell Proliferation

Peripheral blood mononuclear cells (PBMC) were isolated from healthy donors (n = 8) (consent signed upon information given at the time of blood donation) by density gradient centrifugation (Ficoll, Cedarlane, Burlington, ON, Canada), as described [15]. In preliminary experiments, total PBMC was stimulated with either 1 µM soluble ZA or 1 µM isopentenyl pyrophosphate (IPP, Millipore Sigma, Milan, Italy) antigen or 2 µg/mL Bre–Ved–ZA added at the onset of the assay followed by the addition of human recombinant interleukin 2 (IL-2, Peprotech, 30 UI/10 ng/mL final concentration) after 24 h. Afterward, T-lymphocytes and monocytes (Mo) were purified from PBMC using the two specific negative separation kits (Rosettesep, StemCell Technologies, Vancouver, BC, Canada), leading to >98% pure CD2^+^CD3^+^ T-cell or CD14^+^ Mo populations and incubated overnight in RPMI 1640 (supplemented with 10% FBS, Penicillin/Streptomycin and L-Glutamin, all from Gibco, Thermo Fisher, Waltham, MA, USA).

T-lymphocytes were added to each HL-cell line in the absence or presence of Mo (T:Mo ratio 10:1, 10^5^ T-cells: 10^4^ Mo) previously seeded in 96 well U-bottom plates (Sarstedt, Nümbrecht, Germany) in RPMI1640 complete medium as above. The ratio between T-lymphocytes and HL-cells, counted by LUNA, was 10:1 (10^5^ T-cells: 10^4^ HL-cells) with 2.0 µg/mL of Bre–Ved or Bre–Ved–ZA or 1 µM ZA. After 24 h, IL-2 (Peprotech, 30 UI/10 ng/mL final concentration) was added, and co-cultures were continued at 37 °C in a humidified 5% CO_2_ incubator.

The percentage of Vδ2 T-cells was evaluated during culture (at 7, 14, 21 days) by polychromatic immunofluorescence with the anti-Vδ2 γδ123R3 (IgG1) and the anti-CD3 JT3A 289/11/F10 (IgG2a) mAb, followed by AlexaFluor647-goat anti-mouse (GAM) IgG1 and PE-GAM IgG2a (Thermo Fisher). Some samples were stained with the FITC-conjugated anti-Vδ2 REA771 (Miltenyi Biotec, Bergish Gladbach, Germany), the PE-anti-CD45RA MEM56 (Exbio Praha, Vestec, CK), and the APC-anti-CD27 (clone O323, Biolegend, San Diego, CA, USA) and analyzed by flow cytometry [9,10]. Images of T/HL-cell co-cultures were taken using the CELLCYTE X^TM^ imaging recorder (Cytena, Breisgau, Germany).

### 2.5. Cytotoxicity Assays

The direct effect of Bre–Ved–ZA compared to Bre–Ved, both at 20–2.0–0.2 µg/mL, on the different HL-cell lines (KMH2 and L428) was tested with the fluorescent assay C.LIVE Tox Green. Some samples were analyzed over time—up to 6 days—by CELLCYTE X^TM^ (Cytena, Breisgau, Germany), recording images in bright fields and green fluorescence to identify dead cells. Also, the lymphocyte-mediated cytotoxicity was evaluated after the cell cultures of lymphocytes and HL-cells were incubated for 24 h at 37 °C at different effector (lymphocyte) to target (HL-cells) ratios (E:T of 20:1, 10:1, 5:1). Afterwards, the cell cultures were harvested and labeled with the anti-CD2 antibody (γδ54, IgG1) followed by the AlexaFluor647 anti-mouse isotype antiserum to specifically identify lymphocytes. HL-cells were selected as CD2 negative and dead HL-cells were demonstrated with a C.LIVe Tox green probe. Results are shown as percentage CD2 negative C.LIVE Tox green+ HL-cells analyzed on the Cytoflex flow cytometer.

### 2.6. Statistical Analysis

Statistical analysis was performed using a two-tailed unpaired Student’s *t*-test, with Welch correction, using the GraphPad Prism software 5.0. Results are presented as mean ± SD. The cut-off value of significance is indicated in the figure legend.

## 3. Results

### 3.1. Production, Chemical Characterization, and Reactivity of Bre–Ved–ZA ADC

ZA conjugation with Bre–Ved was produced by the reactions used for linking nucleic acids and proteins, generating phosporamidate [23,24,25,26] and resulting in the covalent link between the aminobisphosphonate and the antibody. Figure 1 depicts the mass spectra of Bre–Ved–ZA and unconjugated native Bre–Ved ADC: both spectra show multicharged species, including the singly, doubly, and triply charged species at about *m*/*z* 150 kDa, 75 kDa, and 50 kDa, respectively. The covalent linkage was supported by the increase in the mass of the anti-CD30 antibody loaded with ZA compared to the native sample. Indeed, the monocharged species for native Bre–Ved was detected at 124.78 kDa, while for Bre–Ved–ZA ADC was found at 125.48 kDa (Figure 1, upper vs. lower panel), with an increase in the MW of about 700 units. To evaluate the drug:antibody ratio (DAR) relative to the linkage of ZA to Bre–Ved, LiqC–MS on reduced Bre–Ved–ZA was performed, and the consequent analysis of the molecular peaks revealed that ZA was linked at a mean DAR of approximately 0.5 while vedotin was linked at a mean DAR of 4.0 (Figure 2 and Table 1). ICP–MS showed the measure of phosphorus (ppm of P: 30 ± 1), identifying ZA, compared to that of sulfur (ppm of S: 45 ± 1), representative of Bre–Ved present in the Bre–Ved–ZA preparation. The MALDI–MS, LiqC–MS, and ICP–MS analyses indicate the efficient covalent conjugation of ZA to Bre–Ved, resulting in the Bre–Ved–ZA.

The evidence for a preserved reactivity of Bre–Ved–ZA, compared to the unconjugated Bre–Ved, is provided by immunofluorescence and FACS analysis and depicted in Figure 3A (single histograms) and B (data as mean fluorescence intensity, MFI). The two compounds, incubated at serial dilutions with the CD30^+^ cell lines KMH2, L428, L540, HS445, and RPMI6666, show similar reactivity, optimal at 2.0 µg/mL/10^6^ cells, where Bre–Ved–ZA ADC was even brighter than the unconjugated Bre–Ved (Figure 3B MFI). These data indicate that the ability to recognize CD30 on HL-cells is not altered by ZA conjugation, and Bre–Ved–ZA ADC shows a similar reactivity to Bre–Ved.

### 3.2. Bre–Ved–ZA ADC Localizes in the Lysosomal Compartment and Interferes with HL-Cell Growth

To check the intracellular localization of Bre–Ved–ZA vs. Bre–Ved, all HL-cell lines were primed with 2.0 µg/mL Bre–Ved–ZA ADC for 24 h at 37 °C to allow the reaction of the therapeutic anti-CD30 ADC with the CD30 molecule and the enter the cell of the ADC-CD30 complex. Afterward, cell samples were labeled with anti-LAMP1 mAb (as a lysosomal marker, green) and Syto16 (nuclear marker, blue). Samples were then observed by confocal laser scanning microscopy and analyzed with FluoView 4.3b software. Results depicted in pseudocolors and merged images in Figure 4 show the colocalization of Bre–Ved–ZA (Figure 4A, enlarged in Figure 4B) and Bre–Ved (Figure 4C) with LAMP1 in the lysosomal compartment, confirming that the ADCs undergo the pathway usually followed by internalized therapeutic mAbs [16,17,18,20,21,22].

The evaluation of ATP content in KMH2, L428, L540, HS445, or RPMI6666 upon exposure to serial dilution of Bre–Ved–ZA or Bre–Ved (20, 2.0, 0.2 μg/mL) indicates a significant decrease in cell viability and proliferation in all the cell lines tested. Bre–Ved–ZA was particularly efficient, being active at 0.2 μg/mL (Figure 5A), unlike Bre–Ved. It is worth noting that Caspase 3/7 activation, because of Bre–Ved–ZA or Bre–Ved treatment, was documented at 72 h in KMH2 and RPMI6666 cell lines (Figure 5B). In parallel samples, apoptotic cells were checked by annexin-V labeling by flow cytometry to show the exposure of phosphatidyl-serine on the outer side of the plasma membrane. Apoptosis was differentiated from necrosis by PI staining of non-permeabilized cells, gating annexin-V^+^ PI^+^ cells. Figure 5C shows an example of 25%, 15%, and 7% V^+^PI^+^ apoptotic KMH2 cells after 72 h exposure to 20, 2.0, and 0.2 Bre–Ved–ZA, respectively. Apoptosis triggering was confirmed as nuclear shrinking and fragmentation, observed by confocal microscopy in a bright field and upon nuclear staining with Syto16 (Figure 5D, KMH2, and Figure 5E higher magnification).

The effects of Bre–Ved–ZA on HL-cell proliferation were also quantified by the image analysis at 24, 48, and 72 h of cell cultures, with or without Bre–Ved–ZA or Bre–Ved, at 0.2–2 µg/mL dilution (Figure 6A, KMH2, one representative well/time point; Appendix A shows 4 wells/time point, upper images). The cell area expressed as mm^2^ is depicted in Figure 6B, where it is evident that the growing clusters of KMH2 cells, even after 48 h, are significantly smaller in the presence of Bre–Ved–ZA, which is more efficient than unconjugated Bre–Ved even at very low concentrations (0.2 μg/mL). Similar results were obtained on L428 and L540 cell lines (Figure 6C, Appendix A upper panels, respectively).

Bre–Ved–ZA induced KMH2 cell death, visualized by the increased fluorescence of cells stained by the C.LIVE Tox green probe, as shown in Figure 7A (one representative well/time point). This effect, measured as fluorescence intensity (a.u.), was detectable already at 48 h, and the Bre–Ved–ZA ADC was more efficient than unconjugated Bre–Ved, at the lower concentration (0.2 µg/mL, Figure 7B and Appendix A lower panels, 4 well/time points). Again, superimposable results were collected using L428 and L540 cell lines (Figure 7C, Appendix A lower panels, respectively).

### 3.3. Bre–Ved–ZA ADC Can Stimulate Effector Vδ2 T-Cells with Antitumor Activity

To determine whether the ZA linked to Bre–Ved could trigger Vδ2 T-cell stimulation, an optimal concentration of Bre–Ved–ZA and Bre–Ved impairing the proliferation of HL-cells was selected. This experiment was performed in vitro to mimic the treatment with therapeutic antibodies in a clinical setting. The titration of Bre–Ved–ZA and Bre–Ved was checked on KMH2, L428, and L540 cell line proliferation. The concentration of 2.0 µg/mL was chosen as it could inhibit almost completely the HL-cell growth up to 21 days, while at 0.2 µg/mL, Bre–Ved–ZA was effective only in the first 14 days of culture (effect observed with KMH2 is shown in Figure 8A). Afterward, to assess whether Bre–Ved–ZA ADC can induce the activation of Vδ2 T-cells via ZA delivery, purified T-lymphocytes (>95% pure CD2^+^CD3^+^) were added to monocytes, in the absence or presence of HL-cells, to mimic the tumor microenvironment, at the ratio of 10:1 (10^5^ T-cells: 10^4^ HL-cells and 10^4^ Mo), with 2 µg/mL of Bre–Ved–ZA or Bre–Ved and IL-2 (30 IU/10 ng/mL) was added after 24 h of culture. Parallel cultures with 1 μg/mL of ZA were also set up for comparison to obtain the maximal expansion of Vδ2 T-cells as reported [9,10,22]. Cultures were analyzed at 7, 14, and 21 days by double-immunofluorescence to identify Vδ2 T-lymphocytes using the anti-TCR Vδ2 γδ123R3 and the anti-CD3 JT3A mAbs. FACS analysis showed that the Bre–Ved–ZA ADC could efficiently increase the percentage of Vδ2 T-cells, already evident after 14 d, in contrast with Bre–Ved (Figure 8B as an example, Figure 8C mean ± SD from 5 donors). On day 21, most T-cells in culture were Vδ2 in the presence of Bre–Ved–ZA from each HL-cell cell line, either KMH2 or L428 or L540 (Figure 8C); the effect of the ADC was similar to that obtained with soluble ZA at the concentration of 1 µM (Figure 8C). The imaging of cell cultures shows evident clumps of cells starting from day 7 and at day 14 or 21 upon exposure to 2.0 µg/mL Bre–Ved–ZA ADC (Figure 8D).

To check whether the Vδ2 T-cell population expanded with Bre–Ved–ZA could act as an antitumor effector, we first analyzed the expression of CD45RA and CD27 since this effector (Figure 8E) T-cell subset is reported to bear CD45RA molecule at low levels while missing CD27 [10]. As shown in Figure 8E, at the onset of the cell cultures, the Vδ2-gated T-lymphocytes co-expressed the two molecules as described for naïve cells (upper right plots), although at a lower level of expression compared to ungated T-cells (lower right plots). It is worth noting that the Vδ2 T-cell population obtained after 21 days of culture with Bre–Ved–ZA was mostly CD45RA^+^CD27 (Figure 8F, upper right plot, with still a low expression of CD45RA), as reported for the effector cell subset, unlike Vδ2 T-lymphocytes found with Bre–Ved, which remained double-positive as the Vδ2 T-cells at the onset of culture (Figure 8F, lower right plots). This finding would suggest that Vδ2 T-cells have been triggered after the processing of the ZA linked to the therapeutic anti-CD30 antibody.

Of interest, lymphocytes obtained after 21 days of culture with Bre–Ved–ZA could kill KMH2 or L428 target cells (Figure 8G, red squares) more efficiently than those cultured with Bre–Ved (Figure 8G, gray triangles) or IL-2 alone (Figure 8G, circles). In particular, the cytolytic activity of Bre–Ved–ZA-cultured lymphocyte population against KMH2 cells was still 50% or 40% at 5:1 or 2:1 E.T ratio, respectively vs. 20% or 10% of the other two lymphocyte populations (Figure 8G). Similarly, effector lymphocytes from cultures with Bre–Ved–ZA can kill L428 cells better compared to those obtained with Bre–Ved. These findings suggest that potent antitumor effector Vδ2 T-cells can be generated with the therapeutic anti-CD30 ADC linked to ZA.

## 4. Discussion

We developed a new ADC made of the small molecule zoledronic acid (ZA) and the therapeutic anti-CD30 antibody brentuximab–vedotin (Bre–Ved), able to strongly inhibit the proliferation of CD30^+^ HL/lymphoblastoid tumor cells. This inhibitory effect is about 10-fold stronger than that exerted by the therapeutic antibody Bre–Ved. Also, only Bre–Ved–ZA can trigger the expansion of a subset of potent antitumor T-lymphocytes (Vδ2 T-cells), which can kill some CD30^+^ HL-cell lines efficiently. The CD30 antigen is expressed on HL and some anaplastic lymphomas, and Bre–Ved is a therapeutic tool for these diseases [27,28,29,30,31,32,33,34]. The ADC can hit a molecular target with high selectivity, mainly expressed in tumor cells [35,36,37]. Their specificity is determined by the antibody, while the antitumor effect is associated with the presence of a cytotoxic drug acting on cytoskeleton components or inhibiting enzymes involved in DNA metabolism [35,36,37,38]. Also, other ADCs are composed of an antibody linked to an immune-stimulating cytokine and, for this reason, can show an antitumor effect by triggering killer effector lymphocytes together with tumor-homing [39,40,41,42,43]. In this context, the maytansinoid DM1 microtubular inhibitors have been linked to antibodies against fibronectin domains expressed preferentially in tumors together with IL2 pro-inflammatory cytokine [38]. This ADC can lead to potent anticancer activity in immunocompetent mice models by activation of lymphocytes with tumor cell activity [38]. These ADC or IDC can activate and drive unspecifically IL2-responding killer effector cell populations to the tumor. Herein, we have shown the efficacy of a different approach that can activate a specific anti-HL lymphocyte population composed of Vδ2 T-cells. These cells are a type of lymphocyte effector cells that are intermediate between classical specific cytotoxic T-lymphocytes and innate cells [3,6,12,13]. Indeed, it has been reported that Vδ2 T-cells elicit an immune response independently from the major histocompatibility complex, and they can be triggered through typical activating receptors present in NK cells. Bre–Ved–ZA satisfies these requisites; indeed, this anti-CD30 antibody enables the ADC to target ZA to the CD30^+^ tumor cells, leading to the stimulation of Vδ2 T-lymphocytes.

To conjugate ZA with Bre–Ved, the synthesis of a phosphoramidate bond between the free phosphoric group of ZA and the free amino group of the anti-CD30 antibody has been achieved following the procedure used for linking DNA to peptide [44]. This linkage would make Bre–Ved–ZA ADC similar to the second generation of ADC, as it is stable and processed efficiently by HL-cells.

Anyway, to propose the use of Bre–Ved–ZA ADC to trigger the activation of Vδ2 T-cells against lymphoma cells, this ADC should be further supplemented by an activating cytokine such as IL2 in our experimental system. This limitation could be overcome in the HL microenvironment by the presence of immunostimulating cytokines [27,28,29,30,31,32,33,34].

Interestingly, the Bre–Ved–ZA ADC strongly inhibits the proliferation of CD30^+^ tumor cells, and this effect is accompanied by a stronger increment of apoptosis activating Caspase 3/7 compared to that triggered by Bre–Ved. It is still to be determined which molecular mechanism supported by ZA can favor this increment in the efficacy of Bre–Ved. A possible explanation is that ZA could affect the organization of actin, as reported [45]. Thus, the Bre–Ved–ZA may influence two essential cytoskeleton components, such as microtubules and actin, improving the cytotoxicity of the Bre–Ved.

Also, we have shown that Vδ2 T-cells can be triggered in culture conditions mimicking the tumor microenvironment (TME) present in HL [32,33,34]. Indeed, the combination of T-lymphocytes, monocytes, and HL tumor cells would resemble, at least partially, the HL TME. We have applied a single ratio of 10 unselected T-cells against 1 HL tumor cell with 1 monocyte. Considering that Vδ2 T-cells are usually about 1–3% of unselected T-cells, the ratio between HL-cells and Vδ2 T-cells is near 1-to-1. Actually, Vδ2 T-cells are rare in lymphomas and thus, our experimental conditions may resemble the actual ratio found in HL [46,47]. However, it is well known that in HL, the inflammatory tumor context is composed of several other components, such as mesenchymal stromal cells (MSC), dendritic cells, and macrophages at different stages of differentiation [32,33,34]. Also, the ratio among these cell components may vary from site to site of the affected lymph node, and it is well known that the number of HL tumor cells, such as Reed Sternberg cells, within a lymph node could be low. Altogether, these findings indicate that it is difficult to mimic the HL TME and that our experimental conditions are limited. Anyway, the possibility that Vδ2 T-cells can be expanded in the presence of two components of HL TME (tumor cells and monocyte) could be the evidence that Bre–Ved–ZA can elicit an immune response otherwise absent with Bre–Ved. It is conceivable that Bre–Ved–ZA can enter tumor cells and/or monocytes, leading to small pyrophosphate antigens increase and activation of Vδ2 T-cells. It is still to be proven whether the addition of other cell types, such as MSC, can affect the stimulation exerted by Bre–Ved–ZA. Furthermore, to better mimic the HL TME, a 3D cellular model should be developed to exploit the multiple and spatially dependent interactions among matrix and cellular TME components [32,33,34,36,37,38]. All these points should be further investigated to support better the possible planning of Bre–Ved–ZA in a clinical setting. Finally, the possibility that anti-CD30 antibodies can interact with the CD30 antigen expressed by Vδ2 T-cells, as previously reported [48], was confirmed by immunofluorescence assays (Appendix A). This reactivity may affect the optimal expansion of Vδ2 T-cells, but in our experimental setup, it is evident that the level of reactivity of Bre–Ved and Bre–Ved–ZA antibodies on Vδ2 T-cells is lower than that of tumor cell lines (Appendix A). Also, Bre–Ved–ZA triggered Vδ2 T-cell expansion similarly to the IPP phosphoantigen (Appendix A). Altogether, these findings would suggest that the ADC can enter and exert its cytotoxic effect mainly on tumor cells in co-cultures with lymphocytes.

It is of note that to transfer Bre–Ved–ZA to clinical practice, several challenges should be considered, such as the altered immunogenicity of this novel ADC compared to the native Bre–Ved, as well as the off-target effects [49,50,51,52]. In this context, it has been reported that Bre–Ved can reduce CD30^+^ regulatory T-cells and myeloid-derived suppressor cells (MDSC) [50]. This effect could increase the anti-HL therapeutic action by relieving the immune suppressive TME. Also, the Bre–Ved antibody can have a striking effect by regulating CD30-mediated signaling and its surface expression [50,51,52]. In addition, the influence of the zoledronate linked to anti-CD30 antibodies on bone metabolism should be taken into account [53].

Furthermore, the Bre–Ved–ZA ADC can inhibit CD30+ tumor cell proliferation better, increasing the cytotoxic effect of the Bre–Ved. Taken together, the stronger apoptosis elicited by Bre–Ved–ZA compared to Bre–Ved and the generation of Vδ2 T-cells, it is conceivable that this novel ADC could increase the efficacy of the anti-CD30 immunotherapy. The use of murine models to assess the Bre–Ved–ZA safety profile, as well as the therapeutic advantage compared to Bre–Ved, will be the next step in supporting the treatment of r/r HL with this novel double drug conjugated antibody.

## 5. Conclusions

Herein, we have reported that the Bre–Ved–ZA is a new anti-CD30 ADC bearing the same cellular reactivity as the Bre–Ved, showing strongly increased anti-HL cytotoxic effect together with the ability to stimulate specifically antitumoral Vδ2 T-cells. This last property could be considered the most intriguing feature of this ADC. Indeed, Vδ2 T cells possess functional characteristics of both T and NK lymphocytes, leading to antigen peptide and MHC unspecific immune response. Nevertheless, it appears that the direct antitumor effect of the Bre–Ved antibody is increased by linking with ZA. This would result in the targeting of HL-cells, favoring the development of a stronger double-edged antitumor response triggered by the Bre–Ved–ZA compared to Bre–Ved. In principle, these findings may amplify the therapeutic effect of anti-CD30 antibodies, supporting the use of Bre–Ved–ZA as an additional tool to treat r/r HL.

## Figures and Tables

**Figure 1 cells-13-00862-f001:**
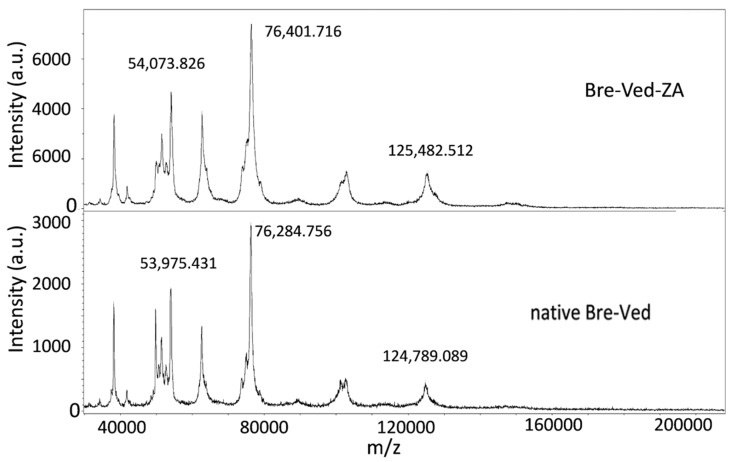
MALDI–MS of Bre–Ved–ZA and Bre–Ved. The plots represent the different molecular weights of the two ADCs. Upper panel: Bre–Ved conjugated with zoledronate; lower panel: native Bre–Ved. The different MWs of the corresponding peaks of the two ADCs indicate the presence of different new molecular species in the preparation of Bre–Ved–ZA due to the covalently linked ZA to the native ADC. The MW of one of these peaks is highlighted to show better the difference between Bre–Ved–ZA and the native Bre–Ved ADC.

**Figure 2 cells-13-00862-f002:**
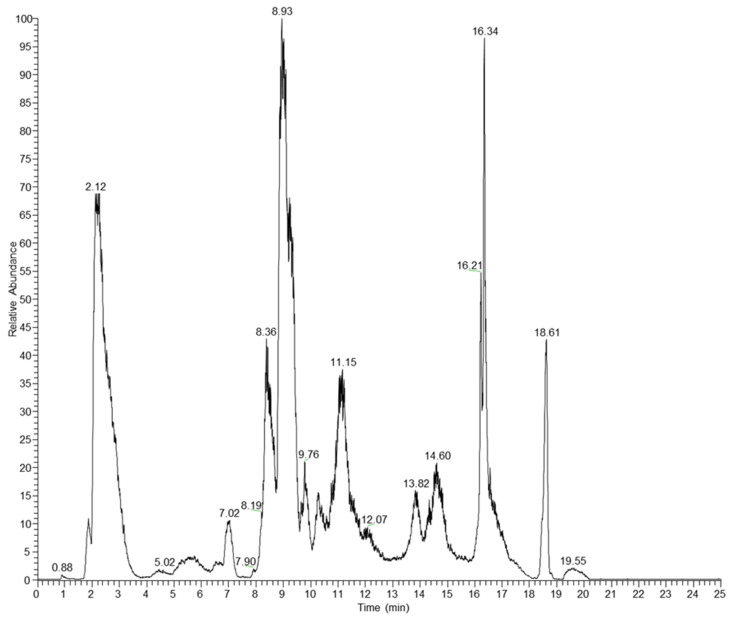
Totally reduced Bre–Ved–ZA LiqC–MS Chromatogram (TIC).

**Figure 3 cells-13-00862-f003:**
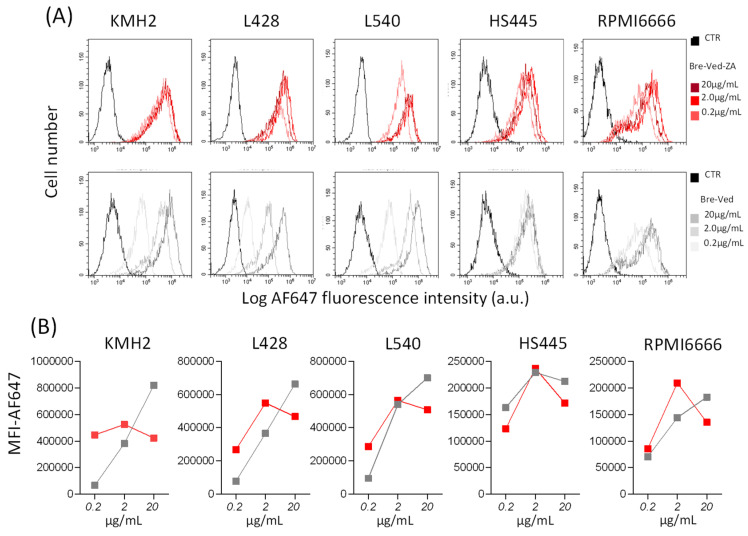
Bre–Ved–ZA ADC reactivity. (**A**) Bre–Ved–ZA titration. Upper panels: the HL-cell lines KMH2, L428, L540, HS445, and RPMI6666 were incubated with 20–0.2 µg/mL/10^6^ cells of Bre–Ved–ZA ADC (upper histograms, red lines) or Bre–Ved (lower histograms, gray lines), followed by the AlexaFluor647-labeled α-hIg antiserum and FACS analysis. Results expressed as Log far red fluorescence intensity (a.u.) vs. the number of cells. (**B**) HL/CD30^+^ lymphoblastoid cell lines were incubated with 20–2.0–0.2 µg/mL/10^6^ cells of Bre–Ved–ZA ADC (red) or Bre–Ved (gray) as in (**B**). Results expressed as mean fluorescence intensity (MFI-AF647, a.u.).

**Figure 4 cells-13-00862-f004:**
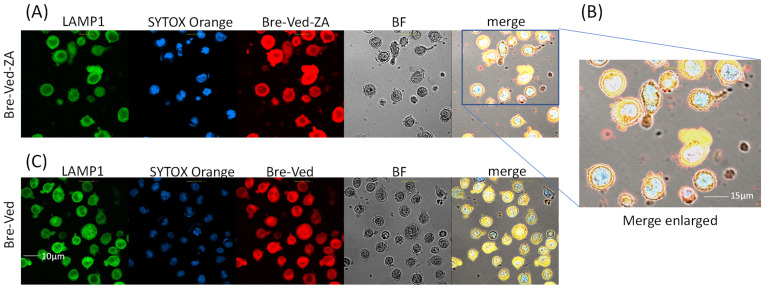
Colocalization of Bre–Ved–ZA and LAMP1 lysosomal marker. The KMH2 cell line was stained with Bre–Ved–ZA ADC (AlexaFluor647, red) (**A**), enlarged in (**B**), or Bre–Ved (**C**) both at 2.0 µg/mL, as indicated, followed by anti-LAMP1 mAb (2.0 µg/mL, AlexaFluor488, green) and SytoX Orange (200 nM, blue) as described in the Methods and Materials section. Samples observed by confocal Laser Scanning Microscopy (400× magnification). To avoid cross-contribution of the various fluorochromes, images have been acquired using the appropriate laser-excitation wavelength with each detector opened at a time. The FluoView 4.3b computer software was used for analysis. Results shown in pseudocolors and merged images are depicted in each panel. Blue: nuclei in pseudocolor. Bar: 10 µm (**A**,**C**), 15 µm (**B**). Results are representative of three independent experiments from three replicated wells.

**Figure 5 cells-13-00862-f005:**
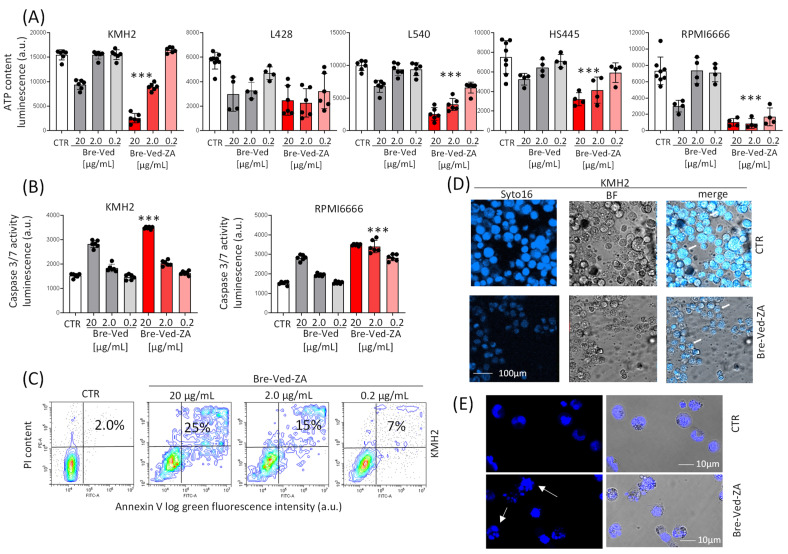
HL-cell-line proliferation and apoptosis upon Bre–Ved–ZA or Bre–Ved treatment. (**A**) ATP content, determined using the CellTiter-Glo^®^ Luminescent Cell Viability Kit, in KMH2, L428, L540, HS445, or RPMI6666 exposed to serial dilution of Bre–Ved or Bre–Ved–ZA (20, 2.0, 0.2 μg/mL) and incubated at 37 °C for 5 days. Luminescence was detected with the VICTORX5 reader and expressed as luminescence arbitrary units (a.u). (**B**) Caspase activation in KMH2 and RPMI6666 cell lines evaluated at 72 h with the luciferase-based Caspase Glo 3/7 3D Assay, expressed as luminescence units (RLU)/5 × 10^4^ cells. Results are the mean ± SD of three independent experiments from 4 replicated wells. *** *p* < 0.001 of Bre–Ved–Za vs. Bre–Ved. (**C**) KMH2 cultured for 72 h with 20 or 2.0 or 0.2 μg/mL Bre–Ved–ZA were labeled with FITC-annexin-V (AV) and propidium iodide (PI) and subjected to FACS analysis. CTR: staining in untreated cells. Apoptotic cells are identified as AV^+^PI^+^. (**D**) Confocal microscopy of KMH2 cells treated for 72 h with 2 μg/mL Bre–Ved–ZA upon nuclear staining with Syto16 (left images). Central images: bright field; right images: merged dark and bright fields. Magnification: 20× objective. Bar: 10 µm. (**E**) Confocal microscopy as in (**D**) at higher magnification (40× objective). Bar: 10 µm. Results are representative of three independent experiments. CTR: KMH2 cultured in medium without antibody. The white arrows in (**D**,**E**) indicate the apoptotic nuclei.

**Figure 6 cells-13-00862-f006:**
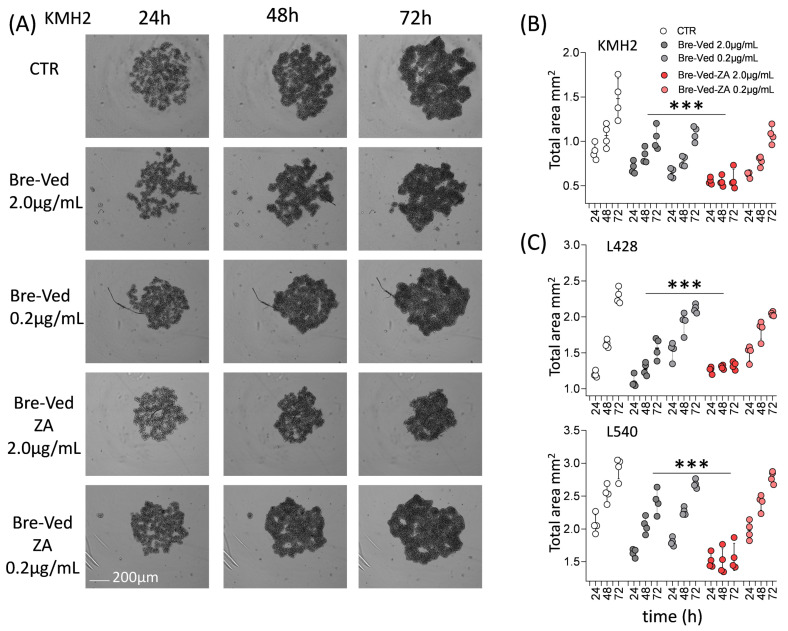
Reduction of HL-cell growth in the presence of Bre–Ved–ZA compared to Bre–Ved. (**A**) Images of KMH2 cell cultures, without or with 2 µg/mL of Bre–Ved–ZA or Bre–Ved, taken at 24, 48, and 72 h with the CELLCYTE X^TM^ imaging recorder are shown. Magnification 50×; one well/each time point is shown. Bar: 100 µm. (**B**) Cell area calculated by image analysis (CELLCYTE Studio software) on images of proliferating KMH2 cell cultures, without or with Bre–Ved–ZA or Bre–Ved (2.0 µg/mL or 0.2 µg/mL) taken at 24, 48, 72 h, expressed as mm^2^. (**C**) Cell area, calculated as in (**A**), of L428 or L540 clusters at 24, 48, and 72 h expressed in mm^2^. Results are the mean ± SD of three independent experiments from 4 replicated wells. *** *p* < 0.001 of Bre–Ved–ZA vs. Bre–Ved at 2.0 µg/mL).

**Figure 7 cells-13-00862-f007:**
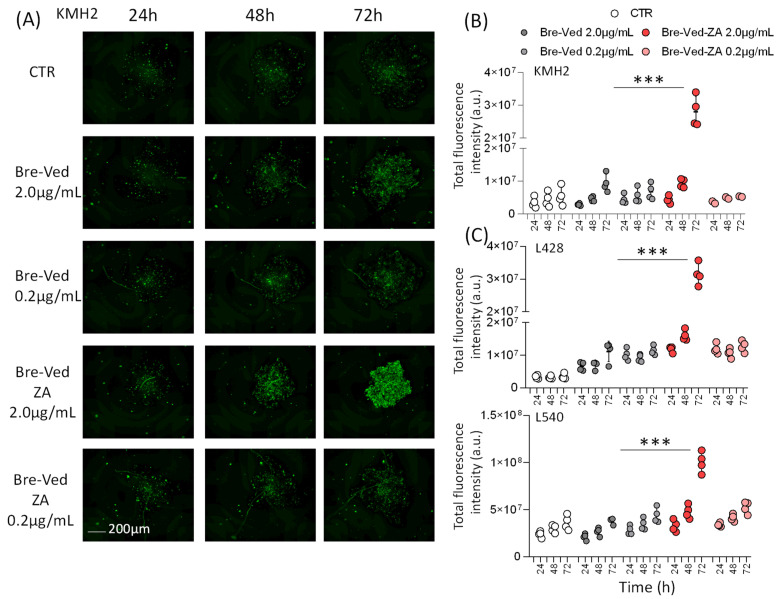
HL-cell death induced by Bre–Ved–ZA ADC compared to Bre–Ved. (**A**) Images of KMH2 cells stained with C.LIVE Tox green fluorescent probe (20 nM) and cultured without or with 2 µg/mL or 0.2 µg/mL Bre–Ved–ZA or Bre–Ved, analyzed at 24, 48, and 72 h with the CELLCYTE X^TM^ imaging recorder. Magnification 50×; one representative well/time point is shown. Bar: 100 µm. (**B**) Fluorescence intensity measured with the CELLCYTE Studio software on images of KMH2 cell cultures, without or with Bre–Ved–ZA or Bre–Ved (2 µg/mL or 0.2 µg/mL) taken at 24, 48, and 72 h, and expressed in arbitrary units (a.u.). (**C**) Fluorescence intensity measured as in A on L428 and L540 cell lines cultured with or without Bre–Ved–ZA or Bre–Ved for the indicated time points (a.u.). Results are the mean ± SD of three independent experiments from 4 replicated wells. *** *p* > 0.001 of Bre–Ved–ZA vs. Bre–Ved at 2.0 µg/mL.

**Figure 8 cells-13-00862-f008:**
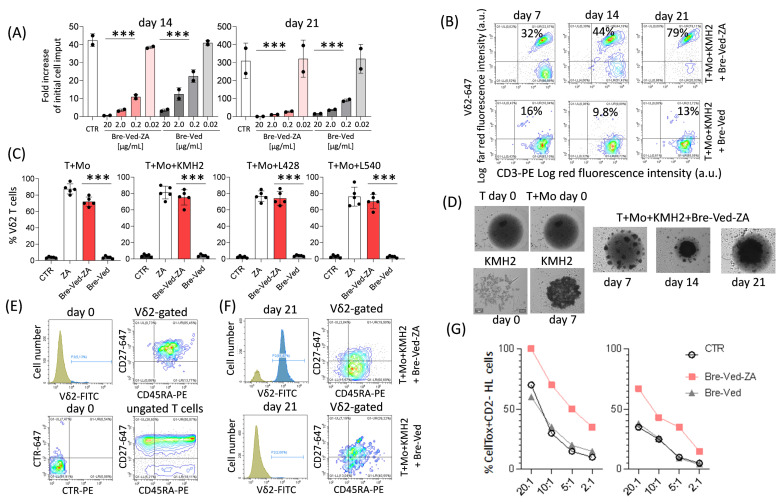
Bre–Ved–ZA ADC induces Vδ2 T-cell proliferation. (**A**) Titration of Bre–Ved–ZA (20, 2.0, 0.2, 0.02 μg/mL) on long-term KMH2 proliferation, determined using the LUNA-II^TM^ automated Cell counter. Cell proliferation in each experimental condition is plotted as a fold increase of the initial input at each time point (days 14 and 21). (**B**) Vδ2 T-cells evaluated by polychromatic immunofluorescence, using the anti-TCR Vδ2 and the anti-CD3 mAbs, at the indicated day (7, 14, and 21) of culture with 2.0 µg/mL Bre–Ved–ZA (upper plots) or Bre–Ved (lower plots). Results expressed as log far red fluorescence intensity vs. log red fluorescence intensity (arbitrary units, a.u.). (**C**) percentage of Vδ2 T-cells after 21 days of culture of purified T-lymphocytes plus Mo and KMH2, or L428 or L540 as indicated, without or with 2.0 µg/mL Bre–Ved or Bre–Ved–ZA or 1 µM ZA. Results are the mean ± SD from 5 lymphocyte donors. (**D**) images of cultures of T-lymphocytes alone (day 0) or with Mo (day 0) or KMH2 only (days 0 and 7) or T+Mo+KMH2 (10^5^ T-cells: 10^4^ KMH2 cells and 10^4^ Mo), taken using the Cell Cyte X imaging recorder on day 7, 14 or 21 of culture incubated at the onset of the assay with 2.0 µg/mL Bre–Ved–ZA. (**E**) Cell samples stained with the FITC-anti-Vδ2, the PE-anti-CD45RA, and the APC-anti-CD27 mAb, analyzed by flow cytometry. Upper panels: staining with anti-Vδ2 mAb on T-cells at day 0 (left) and expression of CD45RA and CD27 on Vδ2-gated T-lymphocytes (right); Lower panels: cells stained with anti-isotype-specific mouse antiserum CTR-PE and CTR-647 (left) or expression of CD45RA and CD27 on ungated T-lymphocytes (right) at time 0. (**F**) Vδ2-gated T-cell population after 21 days of culture (left histograms) with Bre–Ved–ZA (upper panels) or Bre–Ved (lower panels) stained with anti-CD45RA and anti-CD27 mAb (right plots). (**G**) Cells obtained from cultures of T+Mo+KMH2 HL-cells incubated at the onset of culture with Bre–Ved–ZA (red squares) or Bre–Ved (gray triangles) antibodies or IL-2 alone (circles) tested in a cytotoxic assay against KMH2 ((**G**), left) or L428 HL ((**G**), right) target cells at the indicated E:T ratios and analyzed by flow cytometry upon identification of died target cells as CD2 negative and C.LIVE Tox green+ with Cytoflex software. Results are expressed as a percentage of C.LIVE Tox green+CD2^-^ KMH2 (left) or L428 (right) target cells. A representative example of results obtained from three independent experiments with lymphocyte donors is shown. *** = *p* < 0.001.

**Table 1 cells-13-00862-t001:** Molecular species detected in the reduced Bre–Ved–ZA sample by LiqC–MS analysis and conjugation assignment.

	Average Mass (Daltons)	Sum Intensity	Charge State Distribution	Delta Mass	Assignment
LC	23,724	4.47 × 10^7^	10–19	0	LC
	23,898	3.90 × 10^6^	10–19	174	+ZA
	25,040	2.84 × 10^7^	10–17	1316	+Vedotin
	25,214	5.06 × 10^6^	10–18	1490	+Vedotin + ZA
HC	50,321	2.97 × 10^7^	17–37	0	HC glycosylated
	50,483	1.86 × 10^7^	20–34	162	+Hexose
	50,495	4.48 × 10^6^	20–32	174	+ZA
	51,638	3.63 × 10^7^	20–37	1317	+Vedotin
	51,800	2.45 × 10^7^	20–37	1479	+Vedotin + Hexose
	51,812	7.80 × 10^6^	21–43	1491	+Vedotin + ZA
	51,975	6.93 × 10^6^	21–43	1654	+Vedotin + Hexose + ZA
	52,192	5.29 × 10^6^	22–41	1871	+2 Vedotin − 1 MMAE
	52,354	5.54 × 10^6^	21–36	2033	+2 Vedotin − 1 MMAE + Hexose
	52,955	2.30 × 10^7^	20–35	2634	+2 Vedotin
	53,116	1.30 × 10^7^	20–34	2795	+2 Vedotin + Hexose
	53,131	3.80 × 10^6^	20–37	2810	+2 Vedotin + ZA
	53,509	7.11 × 10^6^	24–35	3188	+3 Vedotin − 1 MMAE
	53,671	5.51 × 10^6^	23–38	3350	+3 Vedotin − 1 MMAE + Hexose
	54,271	1.61 × 10^7^	20–39	3950	+3 Vedotin
	54,432	8.81 × 10^6^	20–34	4111	+3 Vedotin + Hexose
	54,444	5.70 × 10^6^	20–33	4123	+3 Vedotin + ZA

The average DAR was calculated based on the following formula: DAR = 2 × (Σ weighted peak area of heavy chain + Σ weighted peak area of light chain)/100. The calculated average DAR_ZA_ was 0.5, while the calculated DAR_ved_ was 4 [24].

## Data Availability

All the homemade reagents and protocol methods used in this study can be available upon motivational requests.

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
