# Peer review of "Antibody–Drug Conjugate Made of Zoledronic Acid and the Anti-CD30 Brentuximab–Vedotin Exert Anti-Lymphoma and Immunostimulating Effects"

_cells, 2024, doi:10.3390/cells13100862_

Round 1
Reviewer 1 Report
Comments and Suggestions for Authors
Very interesting and novel approach to simultaneously target cancer cells by drug-modified antibody and ba activating tumor-reactive gd T cells.
Specific comment: To demonstrate the gd-stimulating activity of the ADC, they used co-cultures of purified T cells with purified monocytes at 1:1 ratio, +/- CD30-positive tumor cells. It would be very informative to include experiments where the gd-stimulating activity of the ADC is tested on PBMC (in comparison to zoledronate and a gd-selective phosphoantigen like HMBPP; of course in presence of exogenous IL-2), and to monitor the expansion of Vg9Vd2 T cells over a period of 1-2 weeks. This could be easily done with 5 donors (also recording the proportion of Vg9Vd2 T cells in the starting population. This is important, because monocytes above a certain threshold (such as perhaps under the 1:1 condition used by the authors) may have a suppressive effect on gd T-cell expansion.
Author Response
Very interesting and novel approach to simultaneously target cancer cells by drug-modified antibody and ba activating tumor-reactive gd T cells.
Specific comment: To demonstrate the gd-stimulating activity of the ADC, they used co-cultures of purified T cells with purified monocytes at 1:1 ratio, +/- CD30-positive tumor cells. It would be very informative to include experiments where the gd-stimulating activity of the ADC is tested on PBMC (in comparison to zoledronate and a gd-selective phosphoantigen like HMBPP; of course, in presence of exogenous IL-2), and to monitor the expansion of Vg9Vd2 T cells over a period of 1-2 weeks. This could be easily done with 5 donors (also recording the proportion of Vg9Vd2 T cells in the starting population. This is important, because monocytes above a certain threshold (such as perhaps under the 1:1 condition used by the authors) may have a suppressive effect on gd T-cell expansion.
Reply
The point raised by the reviewer is of interest. She/he suggested testing the activity of the ADC on PBMC monitoring the expansion of Vδ2 T cells. This because the monocytes above a certain threshold may have suppressive effect on γδT cells expansion. The reviewer intended that under our experimental conditions, a T: Mo ratio of 1:1 has been used. Actually, the ratio between T cells and monocytes was similar to what usually observed in PBMC. Indeed, as indicated on line 204-205 of the original submitted version: “…..T lymphocytes were added to each HL cell line, in the absence or presence of Mo (T:Mo ratio 10:1, 105 T cells: 104 Mo) previously seeded in 96w/U-bottom plates……” and lines 371-372: ….” to mimic the tumor microenvironment, at the ratio of 10:1 (105 T cells: 104 HL cells and 104Mo),”………..
Usually, the amount of monocytes in PBMC, after density gradient separation, is 7-15% while those of T cells (as CD3/TCR+ cells) is 50-75%, thus the ratio among T cells and monocytes is 7-10 to 1. In other words, we tried to mimic the ratio found in the PBMC between T and monocytes without using a higher number of monocytes than what is conceivably found in the PBMC. Also, we eliminated the presence of NK and B cells from PBMC to focus specifically on the interaction among T lymphocytes and monocytes.
Indeed, the reviewer suggested a possible inhibitory effect of monocytes on T cell proliferation associated with a given number of monocytes. This point is of interest because into the tumor microenvironment, monocytes/macrophages can be more frequent than T cells.
However, considering that γδT cells present in PBMC is usually low (1-5%), it is conceivable that in our experimental conditions the number of monocytes is higher than that of γδT cells. Looking at the figure 8C the percentage of γδT cells is less than 5% in all the donors tested compared to 10% of monocytes. This would imply that monocytes did not deliver an inhibitory signal to γδT cells, responding to soluble ZA or ADC linked to ZA, also when more frequent than γδT cells.
Overall, we have to keep in mind that using a human therapeutic antibody Bre-Ved to ZA, the linking of the FC of this antibody with the corresponding FC receptors expressed by monocytes can trigger monocytes themselves. This would favor the generation of monocytes with pro-inflammatory properties instead of regulatory properties. It is still to be defined whether using very high numbers of monocytes (i.e. at a T:Mo ratio of 1:1) could elicit the generation of regulatory monocytes.
Anyway, an additional panel to supplementary figure 5 has been added describing the expansion of γδT cells from PBMC stimulated with the ADC compared to soluble ZA. These data were already available before making experiments with purified T and Mo. Indeed, some preliminary experiments have been performed with unseparated PBMC. Data with the phosphoantigen like HMBPP are not presented as we do not have this compound in the lab and the time to respond to reviewers’ criticism is limited to 10 days. To get the compound would take at least 1 month and additional 15 days to get the results for stimulation. In place of HMBPP, we show some data obtained with another phosphoantigen such as the soluble isopentenyl pyrophosphate (IPP).
From the panel added, it is clear that also PBMC, like co-cultures of T and monocytes, can be triggered by Bre-Ved-ZA ADC similarly or even better than IPP.
Reviewer 2 Report
Comments and Suggestions for Authors
In this interesting work, Brentuximab vedotin was conjugated with zoledronic acid. The authors did several tests to ensure that zoledronic acid was bound to the antibody, Brentuximab vedotin. They also make sure that a such new configurated conjugate did not affect the affinity of brentuximab vedotin to CD30 and that zoledronic acid was transferred into the cells.
They found that, the inhibition of the new antibody-drug-conjugate-zoedronic acid was 10 fold stronger than antibody-drug-conjugate (Brentuximab vedotin) alone.
It introduces a unique ADC that combines Zoledronic Acid with Brentuximab Vedotin showing strong inhibition to proliferation of CD30+ tumor cells but lacks comparison between others ADCs on efficacy and safety profiles. The authors acknowledge experimental difficulties in mimicking tumor microenvironment propose no firm strategies to address such limitation like adding other components or using a.e. advanced 3D cell models.
Authors do not discuss and Challenges of translation their invention into clinical practice such as immunogenicity; off-target effect etc.
Although the present work gives insight into developing new cancer drug by ADC, doing more tests before human trials would make it widely applicable also both its preclinical activities need wide coverage together with how they relate towards achieving long term treatment.
Overall, a good work.
Author Response
In this interesting work, Brentuximab vedotin was conjugated with zoledronic acid. The authors did several tests to ensure that zoledronic acid was bound to the antibody, Brentuximab vedotin. They also make sure that a such new configurated conjugate did not affect the affinity of brentuximab vedotin to CD30 and that zoledronic acid was transferred into the cells.
They found that, the inhibition of the new antibody-drug-conjugate-zoedronic acid was 10 fold stronger than antibody-drug-conjugate (Brentuximab vedotin) alone.
It introduces a unique ADC that combines Zoledronic Acid with Brentuximab Vedotin showing strong inhibition to proliferation of CD30+ tumor cells but lacks comparison between others ADCs on efficacy and safety profiles. The authors acknowledge experimental difficulties in mimicking tumor microenvironment propose no firm strategies to address such limitation like adding other components or using a.e. advanced 3D cell models.
Authors do not discuss and Challenges of translation their invention into clinical practice such as immunogenicity; off-target effect etc.
Although the present work gives insight into developing new cancer drug by ADC, doing more tests before human trials would make it widely applicable also both its preclinical activities need wide coverage together with how they relate towards achieving long term treatment.
Overall, a good work.
Reply
We agree with the reviewer’s criticisms. Indeed, experimental evidence of efficacy and safety profiles would greatly help the further steps to transfer the use of the ADC to the clinic. To consider these points in the Discussion section, some matters raised by this reviewer have been analysed. Indeed, we reported the need to further explore the off-target effects as it has been done for Bre-Ved together the effects on bone metabolism related to zoledronate. Also, it has been proposed to analyse the safety profile in animal murine models. Please see Discussion section lines 508-516 and line 521.
Also, we added some references to support the discussion (new references 50-54).
Round 2
Reviewer 2 Report
Comments and Suggestions for Authors
no more comments